# Purine and Purine Isostere Derivatives of Ferrocene: An Evaluation of ADME, Antitumor and Electrochemical Properties

**DOI:** 10.3390/molecules25071570

**Published:** 2020-03-29

**Authors:** Valentina Rep, Martina Piškor, Helena Šimek, Petra Mišetić, Petra Grbčić, Jasna Padovan, Vesna Gabelica Marković, Dijana Jadreško, Krešimir Pavelić, Sandra Kraljević Pavelić, Silvana Raić-Malić

**Affiliations:** 1Department of Organic Chemistry, Faculty of Chemical Engineering and Technology, University of Zagreb, Zagreb 10000, Croatia; vrep@fkit.hr (V.R.); mpiskor@fkit.hr (M.P.); hsimek@fkit.hr (H.Š.); 2Fidelta d.o.o., Zagreb 10000, Croatia; Petra.Misetic@fidelta.eu (P.M.); Jasna.Padovan@fidelta.eu (J.P.); 3Department of Biotechnology, Center for High-Throughput Technologies, University of Rijeka, Rijeka 51000, Croatia; petra.grbcic@biotech.uniri.hr (P.G.); sandrakp@biotech.uniri.hr (S.K.P.); 4International Relations Office, Faculty of Chemical Engineering and Technology, University of Zagreb, Zagreb 10000, Croatia; vesnagm@fkit.hr; 5Division for Marine and Environmental Research, Ruđer Bošković Institute, Zagreb 10000, Croatia; djadresko@irb.hr; 6Faculty of Medicine, Juraj Dobrila University of Pula, Pula 52100, Croatia; pavelic@unipu.hr

**Keywords:** purine, ferrocene, cytostatic activity, permeability, microsomal stability

## Abstract

Novel purine and purine isosteres containing a ferrocene motif and 4,1-disubstituted (**11a**−**11c**, **12a**−**12c**, **13a**−**13c**, **14a**−**14c**, **15a**−**15c**, **16a**, **23a**−**23c**, **24a**−**24c**, **25a**−**25c**) and 1,4-disubstituted (**34a**−**34c** and **35a**−**35c**) 1,2,3-triazole rings were synthesized. The most potent cytotoxic effect on colorectal adenocarcinoma (SW620) was exerted by the 6-chloro-7-deazapurine **11c** (IC_50_ = 9.07 µM), 6-chloropurine **13a** (IC_50_ = 14.38 µM) and **15b** (IC_50_ = 15.50 µM) ferrocenylalkyl derivatives. The *N*-9 isomer of 6-chloropurine **13a** containing ferrocenylmethylene unit showed a favourable in vitro physicochemical and ADME properties including high solubility, moderate permeability and good metabolic stability in human liver microsomes.

## 1. Introduction

Nucleoside analogues were among the first chemotherapeutic agents that were introduced for the medical treatment of cancer [1,2]. Cytotoxic nucleobase-derived compounds have gained a lot of importance in recent years for combating cancer of different types, usually in combination with other agents [3,4,5]. They were shown to interact with various biological targets, such as cellular kinases, ribonucleotide reductase, and cellular polymerases in order to produce cytotoxic effects [6,7]. The interest in metal complexes of ferrocene-based ligands is due to their favourable physicochemical properties and reversible redox properties that enable ferrocene derivatives to be used as excellent candidates for drug development [8,9]. The antitumor activity of ferrocene-containing compounds is shown to be related to cytotoxic ferrocenium cation and generation of reactive oxygen species (ROS), especially hydroxyl radicals that can be produced under oxidizing conditions through a Fenton-type reaction, whose presence may lead to the damage of the cells [10,11]. To enhance the cytotoxic activity, the ferrocenyl moiety has been successfully incorporated into molecules that have biological activity [12]. In this regard, ferrocifen, obtained by replacing the phenyl group in tamoxifen with ferrocene, exhibits a strong effect against both hormone-dependent (ERα^+^) and hormone-independent (ERα^−^) breast cancer cell types in contrast to tamoxifen, which is only active against ERα^+^ cells [13,14]. Furthermore, the antimalarial drug ferroquine was recently found to enhance anticancer activity of several chemotherapeutics suggesting its potential application as an adjuvant to existing anticancer therapy [15]. It negatively regulates hypoxia-inducible factor-1α (HIF-1α) and showed to be effective under starved and hypoxic conditions frequently observed in advanced solid cancers. The anticancer potential of organometallic nucleoside analogues was explored [16,17,18] and showed that nucleoside analogues of ferrocene exhibited potent antineoplastic activity [19,20,21,22]. Combined application of ferrocenylalkyl nucleobase with anticancer drug cyclophosphamide demonstrated therapeutic synergism of antitumor activity against Lewis lung carcinoma (LLC) [23]. Besides, ferrocene derivatives of thieno[2,3-*d*]pyrimidine, as purine isosteres, exhibited potent anticancer effects in several breast cancer and AML (acute myeloid leukemia) cell lines, despite a loss of mitogen-activated protein kinase-interacting kinase (MNK) potency [24]. On the other hand, ferrocenes incorporating cinchona or carbohydrate moiety through 1,2,3-triazole linker showed significant cytostatic activity [25,26]. Some ferrocene-cinchona hybrids increased reactive oxygen species (ROS) production and induced mitochondrial damage in human multidrug resistant (MDR) cancer cells [27].

Taking into consideration the biological relevance of nucleoside analogues of ferrocene, here we have described the synthesis of novel ferrocene-tagged purine and purine isosteres with the aim to evaluate their antiproliferative activity. Preliminary ADME properties and electrochemical oxidation potential of compounds that exhibited best growth-inhibitory activity were also assessed.

## 2. Results and Discussion

### 2.1. Chemistry

The synthesis of novel purine and purine isosteres (pyrrolo[2,3-*d*]pyrimidine, benzimidazole and indole) containing a ferrocene *N*-1-substituted 1,2,3-triazole moiety (**11a**−**11c**, **12a**−**12c**, **13a**−**13c**, **14a**−**14c**, **15a**−**15c**, **16a**, **23a**−**23c**, **24a**−**24c**, **25a**−**25c**) and a ferrocene 4-substituted 1,2,3-triazole moiety (**34a**−**34c** and **35a**−**35c**) was carried out as shown in Scheme 1, Scheme 2 and Scheme 3.

The key intermediates, the *N*-9- (**5**−**8**) [28,29,30,31] and *N*-7-propargylated purine and 7-deazapurine (**9**−**10**) [30,31] (Scheme 1) and *N*-1-propargylated benzimidazole and indole **20**−**22** (Scheme 2) were obtained by *N*-alkylation of the corresponding heterocyclic base as reported in the literature [32,33]. Target regioselective 1,4-disubstituted 1,2,3-triazoles (**11a**−**11c**, **12a**−**12c**, **13a**−**13c**, **14a**−**14c**, **15a**−**15c**, **16a**, **23a**−**23c**, **24a**−**24c**, **25a**−**25c**) (Scheme 1 and Scheme 2) were prepared by copper(I)-catalysed Huisgen 1,3-dipolar cycloaddition of alkynyl derivatives of corresponding purine and purine isosteres with 1-methylazidoferrocene, 1-azidoethylferrocene and 1-azidoferrocene, which were prepared from ferrocene methanol, 1-hydroxyethylferrocene and 1-bromoferrocene, respectively [34].

The structural diversity of target compounds was further expanded to the synthesis of heterocyclic bases linked through 1-ethyl-1,2,3-triazole to ferrocene (Scheme 3). Reaction of 6-chloro-substituted heterocyclic bases **1** and **3** with the corresponding amines in the presence of KOH under microwave irradiation gave 6-pyrrolidinyl and 6-piperidinyl-substituted intermediates **26**−**29**. *N*-alkylation of the corresponding heterocyclic base **1**, **3**, **26**−**29** with 1,2-dibromoethane in the presence of NaH provided the 2-bromoethyl heterocyclic derivatives **30a**−**30c** [35], **31a**−**31c**. Precursors **30a**−**30c** and **31a**−**31c** were converted in situ, using NaN_3_, to the 2-azidoethyl heterocyclic derivatives **32a**−**32c** [35], **33a**−**33c**. Target purine and 7-deazapurine and ferrocene conjugates **34a**−**34c**, **35a**−**35c** were obtained by a Cu(I)-catalysed click reaction of ethynylferrocene and corresponding *N*-(2-azidoethyl) 6-substituted purine and 7-deazapurine derivatives (Scheme 3).

### 2.2. Biological Profiling

#### 2.2.1. Evaluation of Antiproliferative Activity and Physicochemical Properties

Antiproliferative activity of compounds **11a**−**11c**, **12a**−**12c**, **13a**−**13c**, **14a**−**14c**, **15a**−**15c**, **16a**, **23a**−**23c**, **24a**−**24c**, **25a**−**25c**, **34a**−**34c** and **35a**−**35c** was evaluated in vitro on four human tumor cell lines (American Type Culture Collection, Manassas, VA, USA): metastatic colorectal adenocarcinoma (SW620), ductal pancreatic adenocarcinoma (CFPAC-1), hepatocellular carcinoma (HepG2) and cervical carcinoma (HeLa) as well as normal skin fibroblasts (HFF). The results are presented in Table 1.

Among all tested compounds, 6-chloro-7-deazapurines **11a**−**11c**, 6-chloropurines **13a**−**13c** and ferrocenylalkyl derivatives **15a**−**15c** exhibited the best inhibitory effects, particularly on colorectal adenocarcinoma (SW620) cells. However, compounds that exhibited cytostatic activity were also cytotoxic in the non-tumor HFF cell line. 6-Chloro-7-deazapurine **11c** with ferrocene linked directly to the *N*-1 position of 1,2,3-trazole exhibited the highest activity on SW620 cells (IC_50_ = 9.07 µM). Replacement of the 6-chloro with the 6-amino group in **12a**−**12c** reduced their inhibitory activity in all cell lines. While 6-chloropurine **13a** with the ferrocenylmethylene unit showed better potency compared to its purine isostere **11a**, 6-chloropurine derivatives **13b** and **13c** had lower activity than 7-deazapurine analogues **11b** and **11c**. Similarly, among 2-amino-6-chloropurine and ferrocene conjugates **14a**−**14c**, compound **14a** with the ferrocenylmethylene unit had moderate growth-inhibitory activity, while **14b** with ferrocenylethyl and **14c** with ferrocenyl directly linked to 1,2,3-triazole showed reduced inhibitory effect. Interestingly, among *N*-7 regioisomers of purine-based ferrocene derivatives, only compound **15b** with ferrocenyleth-yl moiety exhibited enhancement of the activity in comparison to its *N*-9 isomer **13b**. Among the benzimidazole, indole and 5-iodoindole derivatives, only benzimidazole **23a** and indole **24a**, both with ferrocenylmethylene unit, had moderate cytostatic activity. Their analogues **23b**, **24b**, **24c** and **25c** containing ferrocenylethyl and ferrocenyl directly linked to the *N*-1 of 1,2,3-triazole showed reduced activity.

6-Chloro-7-deazapurine **34a** with ferrocene connected to the C-4 position of 1,2,3-triazole showed a lack of activity, while its 6-chloropurine structural analogue **35a** displayed only a marginal effect. A comparison of inhibitory activity between 1,4- and 4,1-disubstituted 1,2,3-triazoles showed that a 4,1-disubstituted triazole (as in **11c** and **13c**) is favored over a 1,4-disubstituted triazole (as in **34a** and **35a**). Transformation of 6-chloro to 6-cyclic amine in **34a**, **34c**, **35a** and **35c** did not lead to significant changes in the antiproliferative activity. 6-Pyrrolidinyl (**34b** and **35b**) and 6-piperidinyl-substituted (**35c**) 7-deazapurines and purines were deprived of any activity.

The distribution coefficient Chrom logD, a parameter used to assess the lipophilicity of compounds [36,37], was determined for novel purine and purine isosteres with a ferrocene motif as shown in Table 1. Correlation between lipophilicity and antiproliferative activity was observed and showed that compounds **11a**−**11c**, **13a**, **15a** and **15b** with the highest cytostatic effect had Chrom logD values in the 3.82–5.94 range. A 6-chloro-7-deazapurine moiety increased the lipophilicity of **11a**−**11c** (5.62–5.94) compared to a 6-chloropurine ring of **13a**−**13c** (4.19–4.64). Furthermore, *N*-7 regioisomers **15a**−**15c** and **16a** showed lower Chrom logD values than their *N*-9 regioisomers **13a**−**13c** and **14a**. Indole **24a**−**24c**, 5-iodoindole **25a**−**25c** and ferrocene conjugates that exhibited only marginal inhibition had the highest lipophilicity with Chrom logD values above 6.57. The highest hydrophilicity was determined for adenine-based ferrocene derivatives **12a**−**12c** (Chrom logD = 2.64–3.09) that showed low or did not exhibit any growth-inhibitory effects.

The turbidimetric solubility assay, which allows a rapid determination of the kinetic solubility of newly synthesized compounds using small amounts of their DMSO solutions, was used to assess the kinetic solubility [38]. Results showed that nitrogen heterocycles had the highest impact on solubility (Table 1). Generally, both *N*-9 and *N*-7 isomers of 6-chloropurine had high solubility. Conversely, 6-chloro-7-deazapurine **11a**−**11c**, indole **24a**−**24c** and 5-indole **25a**−**25c** derivatives were less soluble. Among the most active compounds, 6-chloropurine derivatives **13a**, **15a** and **15b** had high solubility, whereas the solubility of 6-chloro-7-deazapurine derivatives **11a**−**11c** was significantly decreased.

#### 2.2.2. Evaluation of Permeability and Metabolic Stability

Compounds **11a**−**11c**, **13a**, **15a** and **15b** with highest cytostatic activities were further screened for permeability and P-glycoprotein (P-gp) substrate assessment, as well as metabolic stability in liver microsomes (human and mouse). These properties have been shown to significantly affect the bioavailability of drugs and to be important in the early drug discovery phases and the lead optimization process [39].

The Madin-Darby canine kidney cells with overexpressed human multidrug-resistant gene (MDCKII-hMDR1) have been frequently used to study bidirectional transport of compounds and assess P-gp substrate potential [40]. Apparent permeability (*P*_app_) values were determined from the amount permeated through the MDCKII-hMDR1 cell monolayer in both the apical-basolateral (AB) and basolateral-apical (BA) direction. *P*_app_ values and the efflux ratios with and without P-gp inhibition with Elacridar are listed in Table 2. 

Compound **11b** is characterized by a high permeability, with an efflux ratio of 1.5 suggesting no influence of P-pg on permeability. All the remaining compounds exhibited a moderate permeability (P_app_A/B), with efflux ratios above 2, classifying them as P-gp substrates, which was further confirmed when they were incubated with a P-gp inhibitor, elacridar. As previously reported in the literature [41], a comparison of efflux ratio generated in the presence and absence of a P-gp inhibitor can be used for the identification of P-gp substrates. A compound is typically considered to be a P-gp substrate when the efflux ratio in the absence of inhibitor is greater than 2 and/or is at least 50% drop is observed in the presence of inhibitor [42]. Among compounds with moderate permeability the 6-chloro-7-deazapurine derivative **11c** with a directly connected ferrocene moiety and *N*-7 regioisomers of 6-chloropurine **15a** and **15b** showed moderate AB permeability, while compounds **11a** and **13a** had moderate to high AB permeability in the range from 8.3 to 8.48 × 10^−6^ cm/s.

Incubation with mouse liver microsomes suggested that that all tested compounds are metabolically unstable with a predicted in vivo clearance above 97% of liver blood flow as summarized in Table 3.

No substantial differences in clearance values for these compounds were observed in mouse microsomes. On the other hand, incubation with human liver microsomes showed the significant impact of type of nitrogen heterocycle on stability. Namely, 6-chloro-7-deazapurine derivatives **11a**−**11c** exhibiting high clearance values (>90% LBF) can be considered as highly labile compounds. On the contrary, 6-chloropurine derivative **13a**, as structural analogue of 6-chloro-7-deazapurine **11a**, with clearance value of 65% LBF showed to be moderately stable. Additionally, *N*-7 regioisomer **15a** had somewhat increased microsomal stability (58% LBF) than its *N*-9 analogue **13a** (65% LBF).

### 2.3. Electrochemical Properties of Selected Compounds by Voltammetric Assays

Compounds **11c**, **13a** and **15b** with strong antiproliferative activity and good physicochemical properties were further evaluated for their electrochemical properties under a wide range of solution conditions. Although the electrochemical behaviour of ferrocene, triazole and their derivatives has been studied in both protic and aprotic solvents [43,44,45,46,47], we have been interested to examine the redox mechanism of the above-mentioned compounds, which may be influenced by the type of nitrogen heterocycle, ferrocene, and alkyl spacer between ferrocene and triazole unit.

The electrochemical properties of 6-chloro-7-deazapurine and 6-chloropurine derivatives of ferrocene **11c**, **13a** and **15b** on a glassy carbon electrode were studied using cyclic (CV) and square-wave voltammetry (SWV) in aqueous electrolyte solutions over a wide pH range. The pH was varied from 2 to 10 in order to determine the effect of pH on the voltammograms (net peak currents and potentials). Figure 1 shows square-wave and cyclic voltammograms for the oxidation of 0.1 mM of **13a** in aqueous buffer electrolytes 3 ≤ pH ≤ 10. The pH of the solution affects the voltammetric response of this compound, i.e., the oxidation of evaluated compounds **11c**, **13a** and **15b** is strongly dependent on it. Furthermore, as can be seen, the best results were obtained at pH 9, so this value was selected for the following experiments.

Figure 2 shows square-wave voltammograms for the oxidation of **11c**, **13a** and **15b** on glassy carbon electrode, in 0.5 mol/L NaClO_4_ at pH 9. The forward (*i*_f_) and backward (*i*_b_) components of the net response (Δ*i* = *i*_f_ − *i*_b_) are shown as well (inset in Figure 2). Two peaks at ~0.4 V (P1) and ~1.1 V (P2) can be observed, indicating that these molecules have two redox active centres. The peak P1 occurs in all supporting electrolytes (i.e., at 2 ≤ pH ≤ 10), while the peak P2 is only observed at pH ≥ 9 (see Figure 1). The potential of peak P1 is pH-independent. Comparing to the literature data, peak P1 can be ascribed to the following charge transfer reaction [Fe^II^(C_5_H_5_)_2_] ⇌ [Fe^III^(C_5_H_5_)_2_]^+^ + e^−^ [44,48]. The forward and backward components of peak P1 are oxidative and reductive currents, respectively (see inset in Figure 2), which indicates that the oxidation of ferrocene is reversible electrode reaction.

Furthermore, in order to clarify the origin of peak P2, ethynyl ferrocene (i) and 6-chloropurine (ii) (bearing no triazole moiety) were analysed as well (under otherwise identical conditions, Figure 3). No anodic peak P2 was detected on SWV responses of these compounds within the investigated potential window. Comparing with the SWV response of **13a** (black curve in Figure 3), it can be concluded that the peak P2 corresponds to the electrode reaction of triazole ring. Both components (*i*_f_ and *i*_b_) of the peak P2 are oxidative currents (see inset in Figure 2), which indicate that the electro-oxidation of 1,2,3-triazole in **11c**, **13a** and **15b** at pH 9 is totally irreversible electrode reaction. The same results were obtained by cyclic voltammetry (under otherwise identical conditions). These observations are in agreement with existing literature on the electrochemical oxidation of triazole-acridine conjugates [49].

In addition, it can be seen from Figure 2 that all investigated compounds provided similar electrochemical responses. More precisely, the electro-oxidation of the 1,2,3-triazole moiety (i.e., peak P2) in **11c**, **13a** and **15b** takes place at almost the same potential. Taking into account the structural difference between **13a** and **15b** in type of the alkyl bridge between the two redox-active centres triazole and ferrocene, it is reasonable to conclude that its effect on the oxidation of the triazole in studied compounds is negligible. However, in the case of **11c**, where the triazole ring is directly attached to the ferrocene, the electrochemical oxidation of ferrocene (i.e., peak P1) occurs at more positive potentials. This result indicates that iron(II) in **11c** is more difficult to oxidize due to the stronger electron-withdrawing effect of the 1,2,3-triazoles ring directly attached to the ferrocene nucleus, or due to its steric hindrance effect on the Fe(II) ion in **11c**.

## 3. Materials and Methods

### 3.1. General Information

All chemicals and solvents were purchased from Aldrich (St. Louis, MO, USA), Fluorochem (Hadfield, UK) and Acros (Geel, Belgium). Anhydrous dimethyl formamide (DMF) was prepared using CaH_2_ and stored over 3Å molecular sieves [50]. Thin layer chromatography (TLC) was performed on pre-coated silica gel 60F-254 plates (Merck, Kenilworth, NJ, USA), while glass column slurry-packed under gravity with 0.063–0.2 mm silica gel (Fluka, Seelze, Germany) was employed for column chromatography. Melting points were determined using a Kofler micro hot-stage (Reichert, Vienna, Austria). ^1^H and ^13^C-NMR spectra were recorded on a Bruker 300 and 600 MHz spectrometers (Bruker, Billerica, MA, USA). All data were recorded in dimethyl sulfoxide (DMSO)-*d*_6_ at 298 K. NMR chemical shifts were referenced to the residual solvent signal of DMSO at δ 2.50 ppm for ^1^H and δ 39.50 ppm for ^13^C. Individual resonances were assigned on the basis of their chemical shifts, signal intensities, multiplicity of resonances, and H–H coupling constants. ^1^H and ^13^C-NMR spectra of compounds are available in Appendix A. Microwave-assisted syntheses were carried out in a microwave oven (Milestone Start S, Sorisole, BG, Italy) at 100 °C and pressure of 1 bar.

4-Chloro-7-(prop-2-yn-1-yl)-7*H*-pyrrolo[2,3-*d*]pyrimidine (**5**) [29], 6-amino-9-(prop-2-yn-1-yl)-9*H*-purine (**6**) [30], 6-chloro-9-(prop-2-yn-1-yl)-9*H*-purine (**7**) [31], 2-amino-6-chloro-9-(prop-2-yn-1-yl)-9*H*-purine (**8**) [32], 6-chloro-9-(prop-2-yn-1-yl)-9*H*-purine (**9**) [31], 2-amino-6-chloro-7-(prop-2-yn-1-yl)-7*H*-purine (**10**) [32], 1-(prop-2-yn-1-yl)-1*H*-benzimidazole (**20**) [34], 1-(prop-2-yn-1-yl)-1*H*-indole (21) [51], 5-iodo-1-(prop-2-yn-1-yl)-1*H*-indole (**22**) [34], 6-(pyrrolidin-1-yl)-9*H*-purine (**28**) [52], 6-(piperidin-1-yl)-*9H*-purine (**29**) [53], and 9-(2-bromoethyl)-6-chloro-*7H*-pyrrolo[2,3-*d*]pyrimidine (**30a**) [36] were synthesized in accordance with procedures given in the literature.

### 3.2. General Procedure for the Synthesis of Purine and Purinomimetics with Ferrocene at N-1 of 1,2,3-Triazole

The corresponding *N*-propargylated heterocyclic base **5**−**10** (1 eq.) was dissolved in methanol, and the corresponding terminal azide (1.2 eq.) and Cu(OAc)_2_ (0.05 eq.) were added. The reaction mixture was stirred at room temperature overnight. The solvent was removed under reduced pressure and the residue was purified by column chromatography (CH_2_Cl_2_:CH_3_OH = 60:1).

*4-Chloro-7-[1-(1-ferrocenymethyl-1,2,3-triazol-4-yl)methyl]-7H-pyrrolo[2,3-d]pyrimidine* (**11a**) Compound **11a** was prepared using the above-mentioned procedure using compound **5** (100 mg, 0.52 mmol) and 1-methylazidoferrocene (150 mg, 0.62 mmol) to obtain **11a** as orange oil (60.3 mg, 33 %). ^1^H-NMR (300 MHz, DMSO-*d*_6_) δ 8.65 (1H, s, H2), 8.06 (1H, s, H5′), 7.80 (1H, d, *J* = 3.6 Hz, H6), 6.67 (1H, d, *J* = 3.6 Hz, H5), 5.56 (2H, s, CH_2_), 5.25 (2H, s, CH_2_), 4.30 (2H, t, *J* = 1.8 Hz, CH-Fc), 4.18–4.14 2H, (m, CH-Fc), 4.12 (5H, s, Cp-Fc). ^13^C-NMR (75 MHz, DMSO-*d*_6_) δ 151.12 (C4), 150.87 (C2), 150.82 (C7a), 143.04 (C4′), 131.77 (C6), 123.51 (C5′), 117.22 (C4a), 99.30 (C5), 82.83 (Cq-Fc), 69.08 (CH-Fc), 69.04 (Cp-Fc), 68.76 (CH-Fc), 49.35 (CH_2_), 40.05 (CH_2_). Anal. calcd. for C_20_H_17_ClFeN_6_: C, 55.52; H, 3.96; N, 19.42. Found: C, 55.46; H, 3.95; N, 19.44.

*4-Chloro-7-[1-(1-ferrocenyl-1-methylmethyl-1,2,3-triazol-4-yl)methyl]-7H-pyrrolo[2,3-d] pyrimidine* (**11b**) Compound **11b** was prepared using the above-mentioned procedure using compound **5** (100 mg, 0.52 mmol) and 1-azidoethylferrocene (159 mg, 0.62 mmol) to obtain **11b** as orange oil (82.6 mg, 35 %). ^1^H-NMR (300 MHz, DMSO-*d*_6_) δ 8.65 (1H, s, H2), 8.09 (1H, s, H5′), 7.80 (1H, d, *J* = 3.6 Hz, H6), 6.67(1H, d, *J* = 3.6 Hz, H5), 5.65 (1H, q, *J* = 7.0 Hz, CH), 5.56 (2H, s, CH_2_), 4.33–4.28 (1H, m, CH-Fc), 4.17–4.15 (2H, m, CH-Fc), 4.08 (5H, H,s, Cp-Fc), 1.78 (3H, d, *J* = 7.0 Hz, CH_3_). ^13^C-NMR (151 MHz, DMSO-*d*_6_) δ 150.60 (C4), 150.36 (C2), 150.30 (C7a), 142.26 (C4′), 131.25 (C6), 121.58 (C5′), 116.70 (C4a), 98.82 (C5), 88.53 (Cq-Fc), 68.61 (Cp-Fc), 68.09 (CH-Fc), 67.62 (CH-Fc), 67.15 (CH-Fc), 66.21 (CH-Fc), 55.60 (CH), 39.54 (CH_2_), 20.83 (CH_3_). Anal. calcd. for C_21_H_19_ClFeN_6_: C, 56.46; H, 4.29; N, 18.81. Found: C, 56.35; H, 4.28; N, 18.87.

*4-Chloro-7-[1-(1-ferrocenyl-1,2,3-triazol-1-yl)methyl]-7H-pyrrolo[2,3-d]pyrimidine* (**11c**) Compound **11c** was prepared using the above-mentioned procedure using compound **5** (100 mg, 0.52 mmol) and 1-azidoferrocene (142 mg, 0.62 mmol) to obtain **11c** as orange oil (39.4 mg, 18 %). ^1^H-NMR (600 MHz, DMSO-*d*_6_) δ 8.68 (1H, s, H2), 8.55 (1H, s, H5′), 7.81 (1H, d, *J* = 3.6 Hz, H6), 6.70 (1H, d, *J* = 3.6 Hz, H5), 5.64 (2H, s, CH_2_), 5.00 (2H, t, *J* = 1.9 Hz, CH-Fc), 4.35–4.31 (2H, m, CH-Fc), 4.18 (5H, s, Cp-Fc). ^13^C-NMR (151 MHz, DMSO-*d*_6_) δ 150.85 (C4), 150.39 (C2), 150.19 (C7a), 142.95 (C4′), 130.83 (C6), 123.33 (C5′), 116.83 (C4a), 99.33 (C5), 93.32 (Cq-Fc), 69.87 (Cp-Fc), 66.57 (CH-Fc), 61.86 (CH-Fc), 34.42 (CH_2_). Anal. calcd. for C_19_H_15_ClFeN_6_: C, 54.51; H, 3.61; N, 20.07. Found: C, 55.40; H, 3.90; N, 20.03.

*6-Amino-9-[1-(1-ferrocenymethyl-1,2,3-triazol-4-yl)methyl]-9H-purine* (**12a**) Compound **12a** was prepared using the above-mentioned procedure using compound **6** (100 mg, 0.58 mmol) and 1-methylazidoferrocene (168 mg, 0.70 mmol) to obtain **12a** as orange powder (132 mg, 76%, m.p. = 107 °C). ^1^H-NMR (300 MHz, DMSO-*d*_6_) δ 8.18 (1H, s, H8), 8.12 (1H, s, H2), 8.07 (1H, s, H5′), 7.21 (2H, s, NH_2_), 5.41 (2H, s, CH_2_), 5.26 (2H, s, CH_2_), 4.30 (2H, dd, *J* = 3.4, 1.6 Hz, CH-Fc), 4.174.14 (2H, m, CH-Fc), 4.13 (5H, s, Cp-Fc). ^13^C-NMR (151 MHz, DMSO-*d*_6_) δ 155.91 (C6), 152.50 (C2), 149.21 (C4), 142.54 (C4′), 140.59 (C8), 123.05 (C5′), 118.51 (C5), 82.35 (Cq-Fc), 68.57 (CH-Fc), 68.56 (Cp-Fc), 68.26 (CH-Fc), 48.85 (CH_2_), 37.96 (CH_2_). Anal. calcd. for C_19_H_18_FeN_8_: C, 55.09; H, 4.38; N, 27.05. Found: C, 54.98; H, 4.39; N, 27.00.

*6-Amino-9-[1-(1-ferrocenyl-1-methylmethyl-1,2,3-triazol-4-yl)methyl]-9H-purine* (**12b**) Compound **12b** was prepared using the above-mentioned procedure using compound **6** (100 mg, 0.58 mmol) and 1-azidoethylferrocene (177 mg, 0.70 mmol) to obtain **12b** as orange powder (53.7 mg, 31 %, m.p. = 223 °C). ^1^H-NMR (600 MHz, DMSO) δ 8.57 (1H, s, H8), 8.26 (1H, s, H5′), 8.15 (1H, s, H2), 7.27 (2H, s, NH_2_), 5.49 (2H, s, CH_2_), 5.01 (2H, s, CH-Fc), 4.33 (2H, s, CH-Fc), 4.19 (5H, s, Cp-Fc). ^13^C-NMR (75 MHz, DMSO) δ 156.46 (C6), 153,17 (C8), 153.01 (C2), 149.87 (C4), 143.39 (C4′), 123.94 (C5′), 119.13 (C5), 93.79 (Cq-Fc), 70.38 (Cp-Fc), 67.08 (CH-Fc), 62.38 (CH-Fc), 38.46 (CH_2_). Anal. calcd. for C_20_H_20_FeN_8_: C, 56.09; H, 4.71; N, 26.16. Found: C, 56.17; H, 4.69; N, 26.11.

*6-Amino-9-[1-(1-ferrocenyl-1,2,3-triazol-1-yl)methyl]-9H-purine* (**12c**) Compound **12c** was prepared using the above-mentioned procedure using compound **6** (100 mg, 0.58 mmol) and 1-azidoferrocene (160 mg, 0.70 mmol) to obtain **12c** as orange powder (13.5 mg, 5.8 %, m.p. > 250 °C). ^1^H-NMR (600 MHz, DMSO) δ 8.57 (1H, s, H8), 8.26 (1H, s, H5′), 8.15 (1H, s, H2), 7.27 (2H, s, NH_2_), 5.49 (2H, s, CH_2_), 5.01 (2H, s, CH-Fc), 4.33 (2H, s, CH-Fc), 4.19 (5H, s, Cp-Fc). ^13^C-NMR (75 MHz, DMSO) δ 156.46 (C6), 153,17 (C8), 153.01 (C2), 149.87 (C4), 143.39 (C4′), 123.94 (C5′), 119.13 (C5), 93.79 (Cq-Fc), 70.38 (Cp-Fc), 67.08 (CH-Fc), 62.38 (CH-Fc), 38.46 (CH_2_). Anal. calcd. for C_18_H_16_FeN_8_: C, 54.02; H, 4.03; N, 28.00. Found: C, 53.86; H, 4.04; N, 27.97.

*6-Chloro-9-[1-(1-ferrocenymethyl-1,2,3-triazol-4-yl)methyl]-9H-purine* (**13a**) Compound **13a** was prepared using the above-mentioned procedure using compound **7** (55 mg, 0.28 mmol) and 1-methylazidoferrocene (81 mg, 0.34 mmol) to obtain **13a** as orange powder (53.2 mg, 43 %, m.p. = 156 °C). ^1^H-NMR (300 MHz, DMSO) δ 8.79 (1H, s, H8), 8.77 (1H, s, H2), 8.15 (1H, s, H5′), 5.60 (2H, s, CH_2_), 5.26 (2H, s, CH_2_), 4.30 (2H, t, *J* = 1.8 Hz, CH-Fc), 4.18–4.15 (2H, m, CH-Fc), 4.13 (5H, s, Cp-Fc). ^13^C-NMR (75 MHz, DMSO) δ 152.15 (C6), 152.13 (C2), 149.53 (C4), 147.92 (C8), 142.18 (C4′), 131.22 (C5), 123.69 (C5′), 82.75 (Cq-Fc), 69.10 (CH-Fc), 69.05 (Cp-Fc), 68.78 (CH-Fc), 49.43 (CH_2_), 39.42 (CH_2_). Anal. calcd. for C_19_H_16_ClFeN_7_: C, 52.62; H, 3.72; N, 22.61. Found: C, 52.46; H, 3.73; N, 22.66.

*6-Chloro-9-[1-(1-ferrocenyl-1-methylmethyl-1,2,3-triazol-4-yl)methyl]-9H-purine* (**13b**) Compound **13b** was prepared using the above-mentioned procedure using compound **7** (100mg, 0.52 mmol) and 1-azidoethylferrocene (157 mg, 0.62 mmol) to obtain **13b** as orange oil (55.5 mg, 23 %). ^1^H-NMR (600 MHz, DMSO) δ 8.79 (1H, s, H8), 8.78 (1H, s, H2), 8.16 (1H, s, H5′), 5.66 (1H, q, *J* = 7.0 Hz, CH), 5.60 (2H, s, CH_2_), 4.31 (1H, s, CH-Fc), 4.17 (2H, d, *J* = 1.7 Hz, CH-Fc), 4.15 (1H, s, CH-Fc), 4.09 (5H, s, Cp-Fc), 1.78 (3H, d, *J* = 7.0 Hz, CH_3_). ^13^C-NMR (75 MHz, DMSO) δ 152.17 (C6), 152.12 (C2), 149.52 (C4), 141.95 (C4′), 131.21 (C5), 122.23 (C5′), 88.99 (Cq-Fc), 69.13 (Cp-Fc), 68.61 (CH-Fc), 68.15 (CH-Fc), 67.67 (CH-Fc), 66.70 (CH-Fc), 56.19 (CH), 39.42 (CH_2_), 21.33 (CH_3_). Anal. calcd. for C_20_H_18_ClFeN_7_: C, 53.65; H, 4.05; N, 21.90. Found: C, 53.54; H, 4.04; N, 21.86.

*6-Chloro-9-[1-(1-ferrocenyl-1,2,3-triazol-1-yl)methyl]-9H-purine* (**13c**) Compound **13c** was prepared using the above-mentioned procedure using compound **7** (100 mg, 0.52 mmol) and 1-azidoferrocene (138 mg, 0. 62 mmol) to obtain **13c** as orange oil (43.2 mg, 19 %). ^1^H-NMR (600 MHz, DMSO) δ 8.85 (1H, s, H8), 8.81 (1H, s, H2), 8.59 (1H, s, H5′), 5.68 (2H, s, CH_2_), 4.99 (2H, s, CH-Fc), 4.33 (2H, s, CH-Fc), 4.19 (5H, s, Cp-Fc). ^13^C-NMR (151 MHz, DMSO) δ 151.77 (C6), 151.65 (C2), 149.05 (C4), 142.18 (C4′), 130.77 (C5), 123.45 (C5′), 93.17 (Cp-Fc), 69.91 (Cp-Fc), 66.61 (CH-Fc), 61.84 (CH-Fc). Anal. calcd. for C_18_H_14_ClFeN_7_: C, 51.52; H, 3.36; N, 23.36. Found: C, 51.57; H, 3.38; N, 23.33.

*2-Amino-6-chloro-9-[1-(1-ferrocenymethyl-1,2,3-triazol-4-yl)methyl]-9H-purine* (**14a**) Compound **14a** was prepared using the above-mentioned procedure using compound **8** (100 mg, 0.48 mmol) and 1-methylazidoferrocene (139 mg, 0.58 mmol) to obtain **14a** as orange powder (123 mg, 57 %; m.p. = 222 °C). ^1^H-NMR (300 MHz, DMSO) δ 8.17 (1H, s, H8), 8.05 (1H, s, H5′), 6.93 (2H, s, NH_2_), 5.33 (2H, s, CH_2_), 5.27 (2H, s, CH_2_), 4.31 (2H, t, *J* = 1.8 Hz, CH-Fc), 4.18–4.15 (2H, m, CH-Fc), 4.14 (5H, s, Cp-Fc). ^13^C-NMR (75 MHz, DMSO) δ 160.32 (C2), 154.33 (C6), 149.85 (C4), 143.46 (C8) 142.74 (C4′), 123.64 (C5), 123.40 (C5′), 82.75 (Cq-Fc), 69.12 (CH-Fc), 69.06 (Cp-Fc), 68.80 (CH-Fc), 49.45 (CH_2_), 38.71 (CH_2_). Anal. calcd. for C_19_H_17_ClFeN_8_: C, 50.86; H, 3.82; N, 24.97. Found: C, 50.91; H, 3.80; N, 24.92.

*2-Amino-6-chloro-9-[1-(1-ferrocenyl-1-methylmethyl-1,2,3-triazol-4-yl)methyl]-9H-purine* (**14b**) Compound **14b** was prepared using the above-mentioned procedure using compound **8** (100 mg, 0.48 mmol) and 1-azidoethylferrocene (147 mg, 0.58 mmol) to obtain **14b** as orange powder (55 mg, 24 %, m.p. = 214 °C). ^1^H-NMR (300 MHz, DMSO) δ 8.17 (1H, s, H8), 8.07 (1H, s, H5′), 6.93 (2H, s, NH_2_), 5.66 (1H, q, *J* = 7.0 Hz, CH), 5.33 (2H, s, CH_2_), 4.32 (2H, d, *J* = 2.1 Hz, CH-Fc), 4.23–4.13 (3H, m, CH-Fc), 4.10 (5H, s, Cp-Fc), 1.80 (3H, d, *J* = 7.0 Hz, CH_3_). ^13^C-NMR (151 MHz, DMSO) δ 159.82 (C2), 153.84 (C6), 149.34 (C4), 142.96 (C8), 141.95 (C4′), 123.15 (C5), 121.47 (C5′), 88.49 (Cq-Fc), 68.64 (Cp-Fc), 68.12 (CH-Fc), 67.66 (CH-Fc), 67.23 (CH-Fc), 66.17 (CH-Fc), 55.70 (CH), 38.24 (CH_2_), 20.77 (CH_3_). Anal. calcd. for C_20_H_19_ClFeN_8_: C, 51.91; H, 4.14; N, 24.22. Found: C, 51.75; H, 4.13; N, 24.24.

*2-Amino-6-chloro-9-[1-(1-ferrocenyl-1,2,3-triazol-1-yl)methyl]-9H-purine* (**14c**) Compound **14c** was prepared using the above-mentioned procedure using compound **8** (100 mg, 0.48 mmol) and 1-azidoferrocene (131 mg, 0.58 mmol) to obtain **14c** as orange powder (44 mg, 21 %, m.p. > 250 °C). ^1^H-NMR (300 MHz, DMSO) δ 8.50 (1H, s, H8), 8.24 (1H, s, H5′), 6.96 (2H, s, NH_2_), 5.41 (2H, s, CH_2_), 4.99 (2H, t, *J* = 1.9 Hz, CH-Fc), 4.384.30 (2H, m, CH-Fc), 4.19 (5H, s, Cp-Fc). ^13^C-NMR (151 MHz, DMSO) δ 159.83 (C2), 153.93 (C6), 149.38 (C4), 142.63 (C4′), 123.28 (C5′), 123.23 (C5), 93.34 (Cq-Fc), 69.89 (Cp-Fc), 66.58 (CH-Fc), 61.86 (CH-Fc), 38.24 (CH_2_). Anal. calcd. for C_18_H_15_ClFeN_8_: C, 49.74; H, 3.48; N, 25.78. Found: C, 49.78; H, 3.47; N, 25,74.

*6-Chloro-7-[1-(1-ferrocenymethyl-1,2,3-triazol-4-yl)methyl]-7H-purine* (**15a**) Compound **15a** was prepared using the above-mentioned procedure using compound **9** (50 mg, 0.26 mmol) and 1-methylazidoferrocene (75 mg, 0.31 mmol) to obtain **15a** as orange powder (56.5 mg, 50 %, m.p. = 96 °C). ^1^H-NMR (300 MHz, DMSO) δ 8.94 (1H, s, H8), 8.80 (1H, s, H2), 8.16 (1H, s, H5′), 5.80 (2H, s, CH_2_), 5.26 (2H, s, CH_2_), 4.30 (2H, t, *J* = 1.8 Hz, CH-Fc), 4.204.14 (2H, m, CH-Fc), 4.12 (5H, s, Cp-Fc). ^13^C-NMR (151 MHz, DMSO) δ 161.57 (C4), 151.74 (C2), 151.17 (C8), 142.54 (C4′), 142.29 (C6), 122.82 (C5′), 121.98 (C5), 82.36 (Cq-Fc), 68.53 (Cp-Fc), 68.51 (CH-Fc), 68.26 (CH-Fc), 48.93 (CH_2_), 41.89 (CH_2_). Anal. calcd. for C_19_H_16_ClFeN_7_: C, 52.62; H, 3.72; N, 22.61. Found: C, 52.56; H, 3.71; N, 22.57.

*6-Chloro-7-[1-(1-ferrocenyl-1-methylmethyl-1,2,3-triazol-4-yl)methyl]-7H-purine* (**15b**) Compound **15b** was prepared using the above-mentioned procedure using compound **9** (90 mg, 0.47 mmol) and 1-azidoethylferrocene (145 mg, 0.56 mmol) to obtain **15b** as orange powder (56.5 mg, 27 %, m.p. = 68 °C). ^1^H-NMR (300 MHz, DMSO) δ 8.94 (1H, s, H8), 8.80 (1H, s, H2), 8.19 (1H, s, H5′), 5.80 (2H, s, CH_2_), 5.66 (1H, q, *J* = 7.0 Hz, CH), 4.334.27 (1H, m, CH-Fc), 4.16 (3H, m, CH-Fc), 4.07 (5H, s, Cp-Fc), 1.78 (3H, d, *J* = 7.0 Hz, CH_3_). ^13^C-NMR (151 MHz, DMSO) δ 161.57 (C4), 151.72 (C2), 142.34 (C4′), 142.30 (C6), 122.02 (C5), 121.37 (C5′), 88.62 (Cq-Fc), 68.59 (Cp-Fc), 68.08 (CH-Fc), 67.61 (CH-Fc), 67.01 (CH-Fc), 66.23 (CH-Fc), 55.70 (CH), 41.97 (CH_2_), 20.94 (CH_3_). Anal. calcd. for C_20_H_18_ClFeN_7_: C, 53.65; H, 4.05; N, 21.90. Found: C, 53.68; H, 4.03; N, 21.86.

*6-Chloro-7-[1-(1-ferrocenyl-1,2,3-triazol-1-yl)methyl]-7H-purine* (**15c**) Compound **15c** was prepared using the above-mentioned procedure using compound **9** (90 mg, 0.47 mmol) and 1-azidoferrocene (129 mg, 0.56 mmol) to obtain **15c** as orange powder (44.7 mg, 22 %, m.p. = 76 °C). ^1^H-NMR (600 MHz, DMSO) δ 9.00 (1H, s, H8), 8.83 (1H, s, H2), 8.58 (1H, s, H5′), 5.87 (2H, s, CH_2_), 4.99 (2H, s, CH-Fc), 4.33 (2H, s, CH-Fc), 4.17 (5H, s, Cp-Fc). ^13^C-NMR (151 MHz, DMSO) δ 161.71 (C4), 151.77 (C2), 151.24 (C8), 143.23 (C6), 142.29 (C4′), 122.75 (C5′), 122.13 (C5), 93.24 (Cq-Fc), 69.89 (Cp-Fc), 66.61 (CH-Fc), 61.74 (CH-Fc), 42.00 (CH_2_). Anal. calcd. for C_18_H_14_ClFeN_7_: C, 51.52; H, 3.36; N, 23.36. Found: C, 51.55; H, 3.34; N, 23.39.

*2-Amino-6-chloro-7-[1-(1-ferrocenymethyl-1,2,3-triazol-4-yl)methyl]-7H-purine* (**16a**) Compound **16a** was prepared using the above-mentioned procedure using compound **10** (70 mg, 0.34 mmol) and 1-methylazidoferrocene (98 mg, 0.41 mmol) to obtain **16a** as orange powder (76 mg, 50 %, m.p. >250 °C). ^1^H-NMR (300 MHz, DMSO) δ 8.49 (1H, s, H8), 8.09 (1H, s, H5′), 6.64 (2H, s, NH_2_), 5.60 (2H, s, CH_2_), 5.26 (2H, s, CH_2_), 4.29 (2H, t, *J* = 1.8 Hz, CH-Fc), 4.174.14 (2H, m, CH-Fc), 4.12 (5H, s, Cp-Fc). ^13^C-NMR (151 MHz, DMSO) δ 164.27 (C4), 159.98 (C2), 142.82 (C6), 142.33 (C4′), 122.74 (C5′), 82.48 (Cq-Fc), 68.54 (CH-Fc), 68.46 (CH-Fc), 68.22 (CH-Fc), 48.89 (CH_2_), 41.56 (CH_2_). Anal. calcd. for C_19_H_17_ClFeN_8_: C, 50.86; H, 3.82; N, 24.97. Found: C, 50.80; H, 3.83; N, 24.91.

### 3.3. General Procedure for the Synthesis of Purinomimetics with Ferrocene at N-1 of 1,2,3-Triazole

The corresponding *N*-propargylated heterocyclic base **20**−**22** (1 eq.) was dissolved in methanol, and the corresponding terminal azide (1.2 eq.) and Cu(OAc)_2_ (0.05 eq.) were added. The reaction mixture was stirred at room temperature overnight. The solvent was removed under reduced pressure and the residue was purified by column chromatography (CH_2_Cl_2_:CH_3_OH = 60:1).

*1-[1-(1-Ferrocenylmethyl-1,2,3-triazol-4-yl)methyl]-1H-benzimidazole* (**23a**) Compound **23a** was prepared using the above-mentioned procedure using compound **20** (70 mg, 0.45 mmol) and 1-methylazidoferrocene (130.2 mg, 0.54 mmol) to obtain **23a** as orange powder (113 mg, 63 %, m.p. = 163 °C). ^1^H-NMR (300 MHz, DMSO) δ 8.32 (1H, s, H2), 8.14 (1H, s, H5′), 7.65 (2H, dd, J = 7.1, 1.9 Hz, H4/7), 7.287.16 (2H,m, H5/6), 5.56 (2H, s, CH_2_), 5.28 (2H, s, CH_2_), 4.354.28 (2H, m, CH-Fc), 4.194.17 (2H, m, CH-Fc), 4.15 (5H, s, Cp-Fc). ^13^C-NMR (151 MHz, DMSO) δ 143.87 (C2), 143.40 (C3a), 142.54 (C4′), 133.45 (C7a), 123.09 (C5′), 122.28 (C6), 121.54 (C4), 119.37 (C5), 110.66 (C7), 82.24 (Cq-Fc), 68.56 (Cp-Fc), 68.29 (CH-Fc), 48.88 (CH_2_). Anal. calcd. for C_21_H_18_FeN_5_: C, 63.65; H, 4.58; N, 17.67. Found: C, 63.71; H, 4.59; N, 17.69.

*1-[1-(1-Ferrocenyl-1-methylmethyl-1,2,3-triazol-4-yl)methyl]-1H-benzimidazole* (**23b**) Compound **23b** was prepared using the above-mentioned procedure using compound **20** (82 mg, 0.53 mmol) and 1-azidoethylferrocene (163.2 mg, 0.64 mmol) to obtain **23b** as brown powder (37.48 mg, 17 %, m.p. = 140 °C). ^1^H-NMR (600 MHz, DMSO) δ 8.33 (1H, s, H2), 8.15 (1H, s, H5′), 7.64 (2H, dd, J = 7.4, 4.7 Hz, H4/), 7.23 (1H, t, J = 7.4 Hz, H6), 7.19 (1H, t, J = 7.5 Hz, H5), 5.66 (1H, q, J = 7.0 Hz, CH), 5.53 (2H, s, CH_2_), 4.31 (1H, s, CH-Fc), 4.214.13 (3H, m, CH-Fc), 4.09 (5H, s, Cp-Fc), 1.78 (3H, d, J = 7.0 Hz, CH_3_). ^13^C-NMR (75 MHz, DMSO) δ 143.94 (C3a), 142.73 (C4′), 133.99 (C7a), 122.77 (C6), 122.19 (C5′), 122.03 (C4), 119.88 (C5), 111.15 (C7), 88.96 (Cq-Fc), 69.13 (Cp-Fc), 68.61 (CH-Fc), 68.15 (CH-Fc), 67.65 (CH-Fc), 66.64 (CH-Fc), 56.12 (CH), 39.86 (CH_2_), 21.37 (CH_3_). Anal. calcd. for C_22_H_20_FeN_5_: C, 64.40; H, 4.91; N, 17.07. Found: C, 64.14; H, 4.92; N, 17.10.

*1-[1-(1-Ferrocenyl-1,2,3-triazol-4-yl)methyl]-1H-benzimidazole* (**23c**) Compound **23c** was prepared using the above-mentioned procedure using compound **20** (83 mg, 0.53 mmol) and 1-azidoferrocene (144.7 mg, 0.64 mmol) to obtain **23c** as orange powder (25 mg, 12 %, m.p. > 250 °C). ^1^H-NMR (600 MHz, DMSO) δ 8.64 (1H, s, H5′), 8.39 (1H, s, H2), 7.66 (2H, s, H4/7), 7.24 (1H, t, J = 7.4 Hz, H6), 7.20 (1H, t, J = 7.2 Hz, H5), 5.62 (2H, s, CH_2_), 4.99 (2H, s, CH-Fc), 4.33 (2H, s, CH-Fc), 4.17 (5H, s, Cp-Fc). ^13^C-NMR (75 MHz, DMSO) δ 144.01 (C3a), 143.39 (C4′), 134.00 (C7a), 124.01 (C5’), 122.84 (C6), 122.12 (C4), 119.96 (C5), 111.15 (C7), 93.78 (Cq-Fc), 70.37 (Cp-Fc), 67.11 (CH-Fc), 62.41 (CH-Fc). Anal. calcd. for C_20_H_17_FeN_5_: C, 62.68; H, 4.47; N, 18.27. Found: C, 62.74; H, 4.46; N, 18.29.

*1-[1-(1-Ferrocenylmethyl-1,2,3-triazol-4-yl)methyl]-1H-indole* (**24a**) Compound **24a** was prepared using the above-mentioned procedure using compound **21** (70 mg, 0.45 mmol) and 1-methylazidoferrocene (130.2 mg, 0.54 mmol) to obtain **24a** as orange powder (25 mg, 14 %, m.p. = 135 °C). ^1^H-NMR (600 MHz, DMSO) δ 7.99 (1H, s, H5′), 7.56 (1H, d, J = 7.9 Hz, H7), 7.51 (1H, d, J = 7.9 Hz, H4), 7.42 (1H, d, J = 3.1 Hz, H2), 7.11 (1H, t, J = 7.2 Hz, H6), 7.00 (1H, t, J = 7.0 Hz,H5), 6.42 (1H, d, J = 2.7 Hz, H3), 5.42 (2H, s, CH_2_), 5.24 (2H, s, CH_2_), 4.28 (2H, t, J = 1.8 Hz, CH-Fc), 4.15 (2H, t, J = 1.8 Hz, CH-Fc), 4.13 (5H, s, Cp-Fc). ^13^C-NMR (101 MHz, DMSO) δ 144.07 (C4′), 135.98 (C7a), 129.04 (C2), 128.68 (C3a), 123.18 (C5′), 121.53 (C6), 120.85 (C4), 119.56 (C5), 110.55 (C7), 101.40 (C3), 82.83 (Cq-Fc), 69.07 (Cp-Fc), 68.77 (CH-Fc), 49.32 (CH_2_), 41.25 (CH_2_). Anal. calcd. for C_22_H_20_FeN_4_: C, 66.68; H, 5.09; N, 14.14. Found: C, 66.48; H, 5.08; N, 14.18.

*1-[1-(1-Ferrocenyl-1-methylmethyl-1,2,3-triazol-4-yl)methyl]-1H-indole* (**24b**) Compound **24b** was prepared using the above-mentioned procedure using compound **21** (75 mg, 0.48 mmol) and 1-azidoethylferrocene (147.9 mg, 0.58 mmol) to obtain **24b** as brown oil (40 mg, 20 %). ^1^H-NMR (300 MHz, DMSO) δ 8.02 (1H, s, H5′), 7.57 (1H, d, J = 7.6 Hz, H7), 7.51 (1H, d, J = 7.8 Hz, H4), 7.41 (1H, d, J = 3.1 Hz, H2), 7.11 (1H, t, J = 7.0 Hz, H6), 7.00 (1H, t, J = 7.0 Hz, H5), 6.42 (1H, dd, J = 3.1, 0.6 Hz, H3), 5.64 (1H, q, J = 7.0 Hz, CH), 5.41 (2H, s, CH_2_), 4.314.27 (1H, m, CH-Fc), 4.174.12 (3H, m, CH-Fc), 4.08 (5H, s, Cp-Fc), 1.77 (3H, d, J = 7.0 Hz, CH_3_). ^13^C-NMR (75 MHz, DMSO) δ 143.75 (C4′), 135.98 (C7a), 128.99 (C2), 128.65 (C3a), 121.79 (C5′), 121.52 (C6), 120.84 (C4), 119.55 (C5), 110.53 (C7), 101.39 (C3), 89.00 (Cq-Fc), 69.14 (Cp-Fc), 68.59 (CH-Fc), 68.13 (CH-Fc), 67.66 (CH-Fc), 66.62 (CH-Fc), 56.02 (CH), 41.25 (CH_2_), 21.37 (CH_3_). Anal. calcd. for C_23_H_22_FeN_4_: C, 67.33; H, 5.40; N, 13.66. Found: C, 67.19; H, 5.41; N, 13.63.

*1-[1-(1-Ferrocenyl-1,2,3-triazol-4-yl)methyl]-1H-indole* (**24c**) Compound **24c** was prepared using the above-mentioned procedure using compound **21** (132 mg, 0. 85 mmol) and 1-azidoferrocene (230.6 mg, 1.02 mmol) to obtain **24c** as brown crystals (32 mg, 9.8 %, m.p. = 120 °C). ^1^H-NMR (300 MHz, DMSO) δ 8.53 (1H, s, H5′), 7.61 (1H, d, J = 8.2 Hz, H7), 7.55 (1H, d, J = 7.8 Hz, H4), 7.48 (1H, d, J = 3.2 Hz, H2), 7.14 (1H, t, J = 7.6 Hz, H6), 7.02 (1H, t, J = 7.0 Hz, H5), 6.47 (1H, d, J = 3.1 Hz, H3), 5.50 (2H, s, CH_2_), 4.97 (2H, t, J = 1.9 Hz, CH-Fc), 4.354.27 (2H, m, CH-Fc), 4.17 (5H, s, Cp-Fc). ^13^C-NMR (75 MHz, DMSO) δ 144.37 (C4′), 136.01 (C7a), 129.14 (C2), 128.77 (C3a), 123.71 (C5′), 121.58 (C6), 120.91 (C4), 119.63 (C5), 110.57 (C7), 101.50 (C3), 93.86 (Cq-Fc), 70.35 (Cp-Fc), 67.06 (CH-Fc), 62.40 (CH-Fc), 41.23 (CH_2_). Anal. calcd. for C_21_H_18_FeN_4_: C, 65.99; H, 4.75; N, 14.66. Found: C, 66.06; H, 4.76; N, 14.63.

*5-Iodo-1-[1-(1-ferrocenylmethyl-1,2,3-triazol-4-yl)methyl]-1H-indole* (**25a**) Compound **25a** was prepared using the above-mentioned procedure using compound **22** (70 mg, 0.25 mmol) and 1-methylazidoferrocene (72.3 mg, 0.30 mmol) to obtain **25a** as brown crystals (79 mg, 15 %, m.p. = 109 °C). ^1^H-NMR (300 MHz, DMSO) δ 7.97 (1H, s, H5′), 7.89 (1H, s, H4), 7.44 (2H, m, H2/7), 7.37 (1H, dd, J = 8.6, 1.5 Hz, H6), 6.39 (1H, d, J = 3.0 Hz, H3), 5.42 (2H, s, CH_2_), 5.24 (2H, s, CH_2_), 4.304.24 (2H, m, CH-Fc), 4.164.14 (2H, m, CH-Fc), 4.12 (5H, s, Cp-Fc). ^13^C-NMR (75 MHz, DMSO) δ 143.77 (C4′), 135.10 (C7a), 131.44 (C3a), 130.17 (C2), 129.43 (C6), 129.25 (C4), 123.19 (C5′), 113.17 (C7), 100.84 (C3), 83.49 (C5), 82.79 (Cq-Fc), 69.06 (Cp-Fc), 68.77 (CH-Fc), 49.32 (CH_2_), 41.35 (CH_2_). Anal. calcd. for C_22_H_19_FeIN_4_: C, 50.60; H, 3.67; N, 10.73. Found: C, 50.65; H, 3.68; N, 10.76.

*5-Iodo-1-[1-(1-ferrocenyl-1-methylmethyl-1,2,3-triazol-4-yl)methyl]-1H-indole* (**25b**) Compound **25b** was prepared using the above-mentioned procedure using compound **22** (223 mg, 0.79 mmol) and 1-azidoethylferrocene (242.3 mg, 0.95 mmol) to obtain **25b** as orange powder (125 mg, 29 %, m.p. = 118 °C). ^1^H-NMR (300 MHz, DMSO) δ 7.97 (1H, s, H5′), 7.89 (1H, s, H4), 7.44 (2H, m, H2/7), 7.37 (1H, dd, J = 8.6, 1.5 Hz, H6), 6.39 (1H, d, J = 3.0 Hz, H3), 5.42 (2H, s, CH_2_), 5.24 (2H, s, CH_2_), 4.304.24 (2H, m, CH-Fc), 4.164.14 (2H, m, CH-Fc), 4.12 (5H, s, Cp-Fc). ^13^C-NMR (75 MHz, DMSO) δ 143.77 (C4′), 135.10 (C7a), 131.44 (C3a), 130.17 (C2), 129.43 (C6), 129.25 (C4), 123.19 (C5′), 113.17 (C7), 100.84 (C3), 83.49 (C5), 82.79 (Cq-Fc), 69.06 (Cp-Fc), 68.77 (CH-Fc), 49.32 (CH_2_), 41.35 (CH_2_). Anal. calcd. for C_23_H_21_FeIN_4_: C, 51.52; H, 3.95; N, 10.45. Found: C, 51.57; H, 3.96; N, 10.49.

*5-Iodo-1-[1-(1-ferrocenyl-1,2,3-triazol-4-yl)methyl]-1H-indole* (**25c**) Compound **25c** was prepared using the above-mentioned procedure using compound **22** (345 mg, 1.23 mmol) and 1-azidoferrocene (334.5 mg, 1.48 mmol) to obtain **25c** as brown crystal (53 mg, 10 %, m.p. = 169 °C). ^1^H-NMR (600 MHz, DMSO) δ 8.52 (1H, s, H5′), 7.92 (1H, s, H4), 7.517.46 (2H, m, H2/7), 7.39 (1H, d, J = 8.6 Hz, H6), 6.44 (1H, d, J = 3.0 Hz, H3), 5.49 (2H, s, CH_2_), 4.97 (2H, s, CH-Fc), 4.32 (2H, s, CH-Fc), 4.17 (5H, s, Cp-Fc). ^13^C-NMR (151 MHz, DMSO) δ 143.60 (C4′), 134.63 (C7a), 131.01 (C3a), 129.77 (C2), 128.97 (C6), 128.81 (C4), 123.22 (C5′), 112.67 (C7), 100.43 (C3), 93.32 (Cq-Fc), 83.09 (C5), 69.86 (Cp-Fc), 66.58 (CH-Fc), 61.91 (CH-Fc), 40.81 (CH_2_). Anal. calcd. for C_21_H_17_FeIN_4_: C, 49.64; H, 3.73; N, 11.03. Found: C, 49.54; H, 3.72; N, 10.99.

### 3.4. General Procedure for N-alkylation of Compounds 26 and 27

To a solution of potassium hydroxide (2 eq.) in water, corresponding heterocyclic base **1**, **3** (1 eq.) and corresponding amine (4 eq.) were added. Reaction mixture was stirred for 10 min under microwave irradiation at 100 °C and 400 W. Formed precipitate was filtered off and dried.

*4-(Pyrrolidin-1-yl)-7H-pyrrolo[2,3-d]pyrimidine* (**26**) Compound **26** was prepared using the above-mentioned procedure using compound 1 (500 mg; 3.26 mmol) to obtain **26** as white powder (613 mg, 95 %, m.p. > 250 °C). ^1^H-NMR (300 MHz, DMSO) δ 11.56 (1H, bs, NH), 8.08 (1H, s, H2), 7.09 (1H, d, J = 3.5 Hz, H6), 6.56 (1H, d, J = 3.5 Hz, H5), 3.71 (4H, bs, CH_2_-pyrrolidine), 1.95 (4H, bs, CH_2_-pyrrolidine). ^13^C-NMR (151 MHz, DMSO) δ 154.77 (C4), 151.18 (C2), 150.89 (7a), 120.31 (C6), 102.43 (4a), 100.79 (C5), 47.38 (CH_2_-pyrrolidine). Anal. calcd. for C_10_H_12_N_4_: C, 63.81; H, 6.43; N, 29.77. Found: C, 63.87; H, 6.44; N, 29.65.

*4-(Piperidin-1-yl)-7H-pyrrolo[2,3-d]pyrimidine* (**27**) Compound **27** was prepared using the above-mentioned procedure using compound 1 (500 mg; 3.26 mmol) to obtain **27** as white powder (659 mg, 84 %, m.p. = 189 °C). ^1^H-NMR (300 MHz, DMSO) δ 11.64 (1H, bs, NH), 8.12 (1H, s, H2), 7.15 (1H, d, *J* = 3.5 Hz, H6), 6.55 (1H, d, *J* = 3.5 Hz, H5), 3.96–3.76 (4H, m, CH_2_-piperidine), 1.73–1.62 (2H, m, CH_2_-piperidine), 1.62–1.50 (4H, m, CH_2_-piperidine). ^13^C-NMR (151 MHz, DMSO) δ 156.27 (C4), 151.90 (C7a), 150.65 (C2), 120.99 (C6), 101.99 (C4a), 100.90 (C5), 46.23 (CH_2_-piperidine), 25.45 (CH_2_-piperidine), 24.25 (CH_2_-piperidine). Anal. calcd. for C_11_H_14_N_4_: C, 65.32; H, 6.98; N, 27.70. Found: C, 65.12; H, 6.99; N, 27.76.

### 3.5. General Procedure for N-alkylation of Compounds 30b, 30c, 31a–31c

To a solution of the corresponding heterocyclic base **1**, **3**, **26**–**29** (1 eq.) in anhydrous DMF, NaH (1.2 eq.) was added and stirred for 30 min under argon atmosphere. 1,2-Dibromoethane (1 eq.) was added and the reaction mixture was stirred at room temperature for 24 h. Solvent was evaporated and the residue was purified by column chromatography (hexane:ethyl acetate = 8:1).

*9-(2-Bromoethyl)-6-(pyrrolidin-1-yl)-7H-pyrrolo[2,3-d]pyrimidine* (**30b**) Compound **30b** was prepared using the above-mentioned procedure using compound **26** (400 mg; 2.13 mmol) to obtain **30b** as yellow powder (106 mg; 16 %, m.p.= 109 °C). ^1^H-NMR (600 MHz, DMSO) δ 8.11 (1H, s, H2), 7.24 (1H, d, J = 3.5 Hz, H8), 6.61 (1H, d, J = 3.5 Hz, H7), 4.52 (2H, t, J = 6.4 Hz, CH_2_), 3.85 (2H, t, J = 6.3 Hz, CH_2_), 3.70 (4H, bs, CH_2_-pyrrolidine), 1.96 (4H, bs, CH_2_-pyrrolidine). ^13^C-NMR (101 MHz, DMSO) δ 155.17 (C6), 151.61 (C2), 150.27 (C4), 124.29 (C8), 103.18 (C5), 101.08 (C7), 47.96 (CH_2_-pyrrolidine), 45.93 (CH_2_), 32.35 (CH_2_). Anal. calcd. for C_12_H_15_BrN_4_: C, 48.83; H, 5.12; N, 18.98. Found: C, 48.73; H, 5.11; N, 18.94.

*9-(2-Bromoethyl)-6-(piperidin-1-yl)-7H-pyrrolo[2,3-d]pyrimidine* (**30c**) Compound **30c** was prepared using the above-mentioned procedure using compound **27** (400 mg; 1.97 mmol) to obtain **30c** as yellow powder (87 mg; 14 %, m.p. = 70 °C). ^1^H-NMR (600 MHz, DMSO) δ 8.14 (1H, s, H2), 7.29 (1H, d, J = 3.6 Hz, H8), 6.59 (1H, d, J = 3.7 Hz, H7), 4.52 (2H, t, J = 6.4 Hz, CH_2_), 3.84 (6H, m, CH_2_, CH_2_-piperidine), 1.65 (2H, m, CH_2_-piperidine), 1.59–1.54 (4H, m, CH_2_-piperidine). ^13^C-NMR (151 MHz, DMSO) δ 156.20 (C6), 150.73 (C4), 150.67 (C2), 124.32 (C8), 102.21 (C5), 100.65 (C7), 46.21 (CH_2_-piperidine), 45.50 (CH_2_), 31.74 (CH*_2_*), 25.46 (CH_2_-piperidine), 24.20 (CH_2_-piperidine). Anal. calcd. for C_13_H_17_BrN_4_: C, 50.50; H, 5.54; N, 18.12. Found: C, 50.60; H, 5.52; N, 18.06.

*9-(2-Bromoethyl)-6-chloro-9H-purine* (**31a**) Compound **31a** was prepared using the above-mentioned procedure using compound **3** (500 mg; 3.23 mmol) to obtain **31a** as white powder (312 mg; 36 %; m.p.= 111°C). ^1^H-NMR (400 MHz, DMSO) δ 8.82 (1H, s, H2), 8.77 (1H, s, H8), 4.75 (2H, t, J = 6.1 Hz, CH_2_), 4.01 (2H, t, J = 6.1 Hz, CH_2_). ^13^C-NMR (101 MHz, DMSO) δ 152.42 (C6), 152.14 (C2), 149.59 (C4), 147.99 (C8), 131.26 (C5), 45.82 (CH_2_), 31.64 (CH_2_). Anal. calcd. for C_7_H_6_BrClN_4_: C, 32.15; H, 2.31; N, 21.42. Found: C, 32.05; H, 2.32; N, 21.47.

*9-(2-Bromoethyl)-6-(pyrrolidin-1-yl)-9H-purine* (**31b**) Compound **31b** was prepared using the above-mentioned procedure using compound **28** (500 mg; 2.65 mmol) to obtain **31b** as white powder (208 mg; 27 %; m.p.= 163 °C). ^1^H-NMR (400 MHz, DMSO) δ 8.22 (1H, s, H2), 8.17 (1H, s, H8), 4.58 (2H, t, J = 6.1 Hz, CH_2_), 4.07 (2H, bs, CH_2_-pyrrolidine), 3.95 (2H, t, J = 6.1 Hz, CH_2_), 3.64 (2H, s, CH_2_-pyrrolidine), 1.95 (4H, s, CH_2_-pyrrolidine). ^13^C-NMR (101 MHz, DMSO) δ 152.95 (C4), 152.70 (C2), 150.37 (C6), 140.71 (C8), 119.81 (C5), 45.07 (CH_2_), 32.03 (CH_2_). Anal. calcd. for C_11_H_14_BrN_5_: C, 44.61; H, 4.76; N, 23.65. Found: C, 44.65; H, 4.75; N, 23.60.

*9-(2-Bromoethyl)-6-(piperidin-1-yl)-9H-purine* (**31c**) Compound **31c** was prepared using the above-mentioned procedure using compound **29** (500 mg; 2.46 mmol) to obtain **31c** as white powder (200 mg; 27 %; m.p. = 149 °C. ^1^H-NMR (400 MHz, DMSO) δ 8.23 (1H, s, H2), 8.20 (1H, s, H8), 4.58 (2H, t, J = 6.1 Hz, CH_2_), 4.20 (4H, bs, CH_2_-piperidine), 3.94 (2H, t, J = 6.1 Hz, CH_2_), 1.73–1.63 (2H, m, CH_2_-piperidine), 1.63–1.55 (4H, m, CH_2_-piperidine). ^13^C-NMR (101 MHz, DMSO) δ 153.58 (C4), 152.39 (C2), 150.95 (C6), 140.18 (C8), 119.34 (C5), 46.08 (CH_2_-piperidine), 45.13 (CH_2_), 31.94 (CH_2_), 26.15 (CH_2_-piperidine), 24.74 (CH_2_-piperidine). Anal. calcd. for C_16_H_16_BrN_5_: C, 46.46; H, 5.20; N, 22.58. Found: C, 46.36; H, 5.19; N, 22.62.

### 3.6. General Procedure for the Synthesis of Azidoethyl Derivatives of Purinomimetics

The corresponding *N*-alkylated heterocyclic base **30a**–**30c**, **31a**–**31c** (1 eq.) was dissolved in acetone. To suspension was added dropwise solution of sodium azide (4 eq.) in water. The reaction mixture was stirred at 60 °C overnight. Solvent was evaporated and crude product was used as is in next step.

### 3.7. General Procedure for the Synthesis of Purinomimetics and Purines with Ferrocene at C-4 of 1,2,3-Triazole

The corresponding terminal azide **32a**–**32c**, **33a**–**33c** (1 eq.) was dissolved in methanol, Cu(OAc)_2_ (0.05 eq), and the ethynylferrocene (1.2 eq.) were added. The reaction mixture was stirred at room temperature overnight. Solvent was evaporated and the residue was purified by column chromatography (CH_2_Cl_2_:CH_3_OH = 60:1).

*4-Chloro-7-(2-(4-ferrocenyl-1,2,3-triazol-1-yl)ethyl)-7H-pyrrolo[2,3-d]pyrimidine* (**34a**). Compound **34a** was prepared using the above-mentioned procedure using compound **32a** (0.76 mmol) and ethynylferrocene (193.2 mg, 0.92 mmol) to obtain **34a** as orange powder (20.7 mg, 6 %, m.p. = 196 °C). ^1^H-NMR (600 MHz, DMSO) δ 8.61 (1H, s, H2), 7.93 (1H, s, H5′), 7.62 (1H, d, J = 3.6 Hz, H6), 6.63 (1H, d, J = 3.6 Hz, H5), 4.88 (2H, dd, J = 6.7, 4.5 Hz, CH_2_), 4.81 (2H, dd, J = 6.7, 4.6 Hz, CH_2_), 4.59 (2H, t, J = 1.8 Hz, CH-Fc), 4.27–4.21 (2H, m, CH-Fc), 3.91 (5H, s,Cp-Fc). ^13^C-NMR (151 MHz, DMSO) δ 150.70 (C4), 150.59 (C7a), 150.27 (C2), 145.16 (C4′), 131.09 (C6), 120.58 (C5′), 116.69 (C4a), 98.76 (C5), 75.60 (Cq-Fc), 69.12 (Cp-Fc), 68.18 (CH-Fc), 66.13 (CH-Fc), 48.80 (CH_2_), 44.54 (CH_2_). Anal. calcd. for C_20_H_17_ClFeN_6_: C, 55.52; H, 3.96; N, 19.42. Found: C, 55.56; H, 3.95; N, 19.37.

*4-(Pyrrolidin-1-yl)-7-[2-(4-ferrocenyl-1,2,3-triazol-1-yl)ethyl]-7H-pyrrolo[2,3-d]pyrimidine* (**34b**) Compound **34b** was prepared using the above-mentioned procedure using compound **32b** (0.68 mmol) and ethynylferrocene (172.2 mg, 0.82 mmol) to obtain **34b** as orange powder (122 mg, 38 %, m.p. = 200 °C). ^1^H-NMR (300 MHz, DMSO) δ 8.12 (1H, s, H2), 7.89 (1H, s, H5′), 6.96 (1H, d, J = 3.6 Hz, H6), 6.53 (1H, d, J = 3.6 Hz, H5), 4.80 (2H, t, J = 5.5 Hz, CH_2_), 4.73–4.64 (2H, m, CH_2_), 4.63–4.59 (2H, m, CH-Fc), 4.29–4.24 (2H, m, CH-Fc), 3.94 (5H, s, Cp-Fc), 3.66 (4H, s, CH_2_-pyrrolidine), 1.91 (4H, s, CH_2_-pyrrolidine). ^13^C-NMR (75 MHz, DMSO) δ 155.19 (C4), 151.71 (C2), 150.31 (C7a), 145.56 (C4′), 123.87 (C6), 121.03 (C5′), 103.11 (C4a), 101.23 (C5), 76.26 (Cq-Fc), 69.64 (Cp-Fc), 68.63 (CH-Fc), 66.63 (CH-Fc), 49.46 (CH_2_), 47.90 (CH_2_-pyrrolidine), 44.27 (CH_2_). Anal. calcd. for C_24_H_25_FeN_7_: C, 61.68; H, 5.39; N, 20.98. Found: C, 61.63; H, 5.38; N, 21.04.

*4-(Piperidin-1-yl)-7-[2-(4-ferrocenyl-1,2,3-triazol-1-yl)ethyl]-7H-pyrrolo[2,3-d]pyrimidine* (**34c**) Compound **34c** was prepared using the above-mentioned procedure using compound **32c** (0.64 mmol) and ethynylferrocene (162.8 mg, 0.77 mmol) to obtain **34c** as orange powder (55 mg, 17 %, m.p. = 171 °C). ^1^H-NMR (600 MHz, DMSO) δ 8.16 (1H, s, H2), 7.89 (1H, s, H5′), 7.03 (1H, d, J = 3.6 Hz, H6), 6.54 (1H, d, J = 3.6 Hz, H5), 4.81 (2H, t, J = 5.8 Hz, CH_2_), 4.67 (2H, t, J = 5.7 Hz, CH_2_), 4.61 (2H, d, J = 1.5 Hz, CH-Fc), 4.28–4.24 (2H, m, CH-Fc), 3.93 (5H, s, Cp-Fc), 3.82–3.79 (4H, m, CH_2_-pipridin), 1.64–1.60 (2H, m, CH_2_-pipridin), 1.52 (4H, d, J = 3.7 Hz, CH_2_-pipridin). ^13^C-NMR (75 MHz, DMSO) δ 156.63 (C4), 151.27 (C7a), 151.20 (C2), 145.60 (C4′), 124.33 (C6), 121.05 (C5′), 102.59 (C4a), 101.40 (C5), 76.22 (Cq-Fc), 69.65 (Cp-Fc), 68.63 (CH-Fc), 66.63 (CH-Fc), 49.41 (CH_2_), 46.63 (CH_2_-piperidine), 44.30 (CH_2_), 25.87 (CH_2_-piperidine), 24.65 (CH_2_-piperidine). Anal. calcd. for C_25_H_27_FeN_7_: C, 62.38; H, 5.65; N, 20.37. Found: C, 62.42; H, 5.64; N, 20.31.

*6-Chloro-9-[2-(4-ferrocenyl-1,2,3-triazol-1-yl)ethyl]-9H-purine* (**35a**) Compound **35a** was prepared using the above-mentioned procedure using compound **33a** (0.76 mmol) and ethynylferrocene (193.2 mg, 0.92 mmol) to obtain **35a** as orange powder (123 mg, 37 %, m.p. = 240 °C).^1^H-NMR (300 MHz, DMSO) δ 8.76 (1H, s, H2), 8.50 (1H, s, H5′), 8.02 (1H, s, H8), 4.94–4.90 (2H, m, CH_2_), 4.84 (2H, d, J = 6.1 Hz, CH_2_), 4.60 (2H, d, J = 1.6 Hz, CH-Fc), 4.28–4.25 (2H, m, CH-Fc), 3.94 (5H, s, Cp-Fc). ^13^C-NMR (75 MHz, DMSO) δ 152.43 (C6), 152.06 (C2), 149.52 (C4), 147.73 (C8), 145.87 (C4′), 131.16 (C5), 121.23 (C5′), 76.02 (Cq-Fc), 69.64 (Cp-Fc), 68.70 (CH-Fc), 66.67 (CH-Fc), 48.88 (CH_2_), 44.37 (CH_2_). Anal. calcd. for C19H16ClFeN7: C, 52.62; H, 3.72; N, 22.61. Found: C, 52.56; H, 3.71; N, 22.66.

*6-(Pyrrolidin-1-yl)-9-[2-(4-ferrocenyl-1,2,3-triazol-1-yl)ethyl]-9H-purine* (**35b**) Compound **35b** was prepared using the above-mentioned procedure using compound **33b** (0.68 mmol) and ethynylferrocene (172.2 mg, 0.82 mmol) to obtain **35b** as orange powder (124 mg, 39 %, m.p. = 230 °C). ^1^H-NMR (300 MHz, DMSO) δ 8.21 (1H, s, H2), 7.96 (1H, s, H5′), 7.84 (1H, s, H8), 4.86 (2H, t, J = 5.6 Hz, CH_2_), 4.70 (2H, t, J = 5.6 Hz, CH_2_), 4.62 (2H, t, J = 1.8 Hz, CH-Fc), 4.31–4.21 (2H, m, CF-Fc), 3.95 (7H, s, Cp-Fc), 3.59 (2H, s, CH_2_-pyrrolidine), 1.89 (4H, s, CH_2_-pyrrolidine). ^13^C-NMR (75 MHz, DMSO) δ 152.90 (C6), 152.72 (C2), 150.36 (C4), 145.73 (C4’), 140.32 (C8), 121.12 (C5’), 119.71 (C5), 76.13 (Cq-Fc), 69.64 (Cp-Fc), 68.65 (CH-Fc), 66.63 (CH-Fc), 48.94 (CH_2_-pyrrolidine), 43.45 (CH_2_-pyrrolidine). Anal. calcd. for C_23_H_24_FeN_8_: C, 58.98; H, 5.17; N, 23.93. Found: C, 58.92; H, 5.16; N, 23.96.

*6-(Piperidin-1-yl)-9-[2-(4-ferrocenyl-1,2,3-triazol-1-yl)ethyl]-9H-purine* (**35c**) Compound **35c** was prepared using the above-mentioned procedure using compound **33c** (0.32 mmol) and ethynylferrocene (162.8 mg, 0.77 mmol) to obtain **35c** as orange powder (47 mg, 15 %, m.p. = 233 °C). ^1^H-NMR (300 MHz, DMSO) δ 8.22 (1H, s, H2), 7.97 (1H, s, H5′), 7.88 (1H, s, H8), 4.87 (2H, t, J = 5.5 Hz, CH_2_), 4.70 (2H, t, J = 5.7 Hz, CH_2_), 4.66–4.55 (2H, m, CH-Fc), 4.30–4.22 (2H, m, CH-Fc), 4.14 (4H, bs, CH_2_-piperidin), 3.94 (5H, s, Cp-Fc), 1.62 (2H, s, CH_2_-piperidine), 1.52 (4H, s, CH_2_-piperidine). ^13^C-NMR (75 MHz, DMSO) δ 153.52 (C6), 152.39 (C2), 150.95 (C4), 145.75 (C4′), 121.14 (C5′), 119.22 (C5), 76.10 (Cq-Fc), 69.65 (Cp-Fc), 68.66 (CH-Fc), 66.64 (CH-Fc), 48.93 (CH_2_), 43.48 (CH_2_), 26.04 (CH_2_-piperidine), 24.68 (CH_2_-piperidine). Anal. calcd. for C_24_H_26_FeN_8_: C, 59.76; H, 5.43; N, 23.23. Found: C, 59.70; H, 5.42; N, 23.28.

### 3.8. Kinetic Solubility Assay

The compounds’ DMSO stock solutions in concentration of 10 mM were serially diluted by factor 3.3×, 3×, 3.3× and 3× in DMSO. Prepared solutions were spiked into phosphate-buffered saline (PBS, 100 mM) to obtain final compounds’ concentrations of 100, 30, 10, 3 and 1 µM (3 µL of prepared dilutions were spiked to 297 µL of PBS, with final DMSO content of 1%, *v*/*v*). All compounds were tested in two replicas per concentration. Prepared solutions were incubated at 37 °C for 2 h with gentle shaking. Blank solutions were also prepared by spiking DMSO in PBS buffer (1% of DMSO, *v*/*v*). After the incubation, absorbance of suspensions was measured by Microplate reader Infinite F500 (Tecan, Männedorf, Switzerland) at 620 nm and compared to a blank control to obtain the tested solutions’ absorbance, which is proportionally increased with concentration of insoluble particles. Standard compounds α-naphtoflavone (low solubility) and sulfaphenazole (high solubility) were used as controls. Compound/control samples are compared to the blank solution. Precipitation occurred if significant increase of sample absorbance is observed, i.e., when its absorbance is 3-fold standard deviation of average blank absorbance. Results were expressed as an estimated solubility range (lower and upper bound).

### 3.9. Chrom logD Determination

The distribution coefficient, Chrom logD, was determined from compounds’ gradient retention times by employing a high-performance liquid chromatography method with fast acetonitrile gradient. Compounds were prepared for analysis in final concentration of 1.25 mM by dilution of 10 mM stock solutions in DMSO with acetonitrile. Aqueous mobile phase was 50 mM ammonium acetate adjusted with ammonia solution to pH 7.4 and the organic mobile phase was acetonitrile. A reversed phase HPLC column Luna C18, 50 × 3 mm i.d., 5 µm particle size (Phenomenex, Torrance, CA, USA) was used for analysis. Sample injection volume was 2 µL and the flow rate was 1 mL min^−1^. All compounds were analysed in duplicates. Fast-gradient elution was as follows: from 0 to 100% of acetonitrile (0−3 min), 100% of acetonitrile (3.0−3.5 min), from 100 to 0% of acetonitrile (3.5−3.7 min), and re-equilibration with 100% of aqueous mobile phase (3.7−5.0 min). All measurements were performed on an HPLC-DAD instrument Agilent 1100 Series (Agilent Technologies, Santa Clara, CA, USA) coupled with a mass spectrometer Micromass Quattro API (Waters, MA, USA). MassLynx software, version 4.1 (Waters, Manchester, UK) was used for data acquisition and processing. Before compound analysis, a calibration step was performed. A set of standard compounds with literature CHI values [36] were plotted against gradient retention times to obtain a calibration equation, which was used to determine CHI value from the compounds’ retention times. Chrom logD values were calculated from the CHI value using the equation Chrom logD = 0.0857 × CHI − 2. [38]

### 3.10. ADME Properties

#### 3.10.1. Permeability and P-glicoprotein Substrate Assessment

MDCKII-hMDR1 cell monolayer was used for determination of permeability and P-gp substrate assessment. MDCKII-hMDR1 cells (Solvo Biotechnology, Szeged, Hungary) were grown in the controlled atmosphere (37 °C, 95% humidity, 5% CO_2_) in MDCK cell culture medium containing Dulbecco’s Modified Eagle Medium (DMEM) + 10% Fetal bovine serum (FBS, inactivated) + 1% glutamax-100 + 1% antibiotic/antimycotic + 1% of Non-essential amino acids Minimum Essential Medium (MEM NEAA). Cultures were split every 3–4 days using 0.05% Trypsin-EDTA. Cells were seeded 4 days prior to the experiment in concentration of 0.3 × 10^6^ cells mL^−1^ and cultured in CO_2_ incubator. On the experiment day, cell monolayers were washed with Dulbecco’s phosphate buffer saline (D-PBS) and equilibrated for 45 min in CO_2_ incubator in D-PBS containing 1% DMSO (1% *v*/*v*) with or without Elacridar (2 µM). Compounds, in final concentration of 10 µM, were prepared in transport medium containing Lucifer yellow (100 µM), with DMSO contentment of 1% (*v*/*v*) and in the presence and without Elacridar. Lucifer yellow calibration curve was prepared by serial dilution (12 points, with 100 µM as the highest point). Monolayer integrity was examined by fluorescent measurement on a Microplate reader Infinite F500 (Tecan) by using an excitation of 485 nm and emission of 530 nm. Experiment was started by applying the solutions containing test compounds to apical and basolateral side of the cell monolayer. Starting compound concentration (C_0_) was sampled at the start of the experiment and apical and basolateral compartments were sampled after incubation at 37 °C with gentle shaking for 60 min. Compound were tested in duplicate. Amprenavir was used as a control with low permeability and as a P-gp substrate without inhibitor presence, turning into a highly permeable compound with the inhibitor present. Diclofenac was used as a control with high permeability and no interaction with P-gp.

Samples were analyzed on an ABSciex API 4000 Triple Quadrupole Mass Spectrometer (Sciex, Division of MDS Inc., Toronto, ON, Canada) coupled to a UHPLC Nexera X2 (Shimadzu, Kyoto, Japan). Samples (1 µL) were injected onto a reversed phase HPLC column Luna Omega Polar C18, 30 × 2.1 mm i.d., 1.6 µm particle size (Phenomenex) which was kept at 50 °C. Aqueous solution was 0.1% formic acid in deionized water and the organic mobile phase was water/acetonitrile/formic acid mixture (90/10/0.1, *v*/*v*/*v*). Flow rate was kept at 0.7 mL min^−1^ for all measurements. Fast-gradient elution was as follows: 2% of organic mobile phase through 0.15 min, from 2 to 95% of organic mobile phase (0.15−0.7 min), 95% of organic mobile phase (0.7−1.1 min), from 95 to 2% of organic mobile phase (1.1−1.11 min), and re-equilibration with 98% of aqueous mobile phase (1.11−1.5 min). Positive ion mode with turbo spray, an ion source temperature of 550 °C and a dwell time of 75 ms were utilized for mass spectrometric detection. Quantitation was performed using multiple reaction monitoring (MRM) at the specific transition corresponding to the compound of interest. Warfarin was used as an internal standard. The ratios between compound and internal standard peak areas were used instead of real compound concentration. Apparent permeability coefficient, P_app_ (nm s^−1^) values were calculated by using the following Equation (1):P_app_ [nm s^−1^] = (dQ/dT) × (1/C_0_) × (1/A)(1)
where: dQ/dT = permeability rate; C_0_ = initial concentration in donor compartment; A = surface area of the cell monolayer (0.7 cm^2^).

The efflux ratio in the presence or absence of the P-gp inhibitor was calculated from P_app_ values, using following Equation (2):Efflux ratio = P_app_ (BA)/P_app_(AB)(2)

#### 3.10.2. Metabolic Stability in Liver Microsomes

Metabolic stability of compounds was assessed in human and mouse liver microsomes (Corning, Tewksbury, MA, USA). Compounds, in final concentration of 1 µM, were incubated in phosphate buffer (50 mM, pH 7.4) for 60 min at 37 °C together with liver microsomes and NADPH generating system (nicotinamide adenine dinucleotide phosphate (NADP, 0.5 mM), glucose-6-phosphate (G6P, 5 mM), glucose-6-phosphate dehydrogenase (1.5 U mL^−1^) and magnesium chloride (0.5 mM)). Also, compounds were incubated without the presence of NADPH cofactor, as a buffer stability control. Metabolic activity of liver microsomes was verified by including testosterone and propranolol as positive controls, as well as caffeine as a negative control. Sampling was performed at six time points (0, 10, 20, 30, 45 and 60 min), followed by reaction termination by addition of acetonitrile/methanol mixture (2:1, *v*/*v*) containing dicofenac as internal standard. Samples were analyzed as previously described for samples from permeability assay. The in vitro half-life (t_1/2_) was calculated from the slope of the linear regression by plotting ln % remaining of parent compound against incubation time. In vitro intrinsic clearance, CL_int_ was calculated from half-life using following Equation (3):(3)$${\mathrm{CL}}_{\mathrm{int}}\;[µL\;\min^{-1\;}{\mathrm{mg}}^{-1}]=\frac{\ln2}{t1/2}\times \frac{\mathrm{mL}\;\mathrm{per}\;\mathrm{incubation}}{\mathrm{mg}\;\mathrm{protein}\;\mathrm{per}\;\mathrm{incubation}}$$
where 52.5 mg protein/g liver is used as a constant.

Predicted in vivo hepatic clearance, in vivo CL_h_ was calculated as follows:(4)$$\mathrm{in}\;\mathrm{vivo}\;{\mathrm{CL}}_{h}\;[\mathrm{mL}\;\min^{-1}\;{\mathrm{kg}}^{-1}]=\frac{\mathrm{CLint}\times (\mathrm{mg}\;\mathrm{protein}/g\;\mathrm{liver})\times Q\times (\mathrm{LW}/\mathrm{BW})\;}{Q+(\mathrm{CLint}\times (\mathrm{mg}\;\mathrm{protein}/g\;\mathrm{liver})\times (\mathrm{LW}/\mathrm{BW}))}$$
where: LW/BW = liver weight/body weight [g kg^−1^]; 25.7 (human), 87.5 (mouse); Q = LBF = liver blood flow [mL min^−1^ kg^−1^]; 21/1.26 (human), 131/7.86 (mouse).

Predicted in vivo hepatic clearance can be expressed as %LBF and calculated as follows:(5)$$\mathrm{in}\;\mathrm{vivo}\;{\mathrm{CL}}_{h}\;[\%\mathrm{LBF}]=\frac{\mathrm{in}\;\mathrm{vivo}\;\mathrm{CLh}}{LBF}\times 100$$

### 3.11. Voltammetric Measurements

All chemicals used in the experiments were of the best grade commercially available (Sigma-Aldrich) and were used without further purification. Stock standard solutions of compounds **11c**, **13a**, **15b** (*c* = 2.0 × 10^−3^ mol/L) and 6-chloropurine (*c* = 1.2 × 10^−2^ mol/L) were prepared from dry pure substances in dimethyl sulfoxide (DMSO, p.a.), purchased from Kemika (Zagreb, Croatia). Stock standard solution of ethynyl ferrocene (*c* = 1.1 × 10^−2^ mol/L) was prepared from dry pure substance in ethanol (Kemika). For the supporting electrolyte, analytical grade NaClO_4_ (Kemika) was used. Water was deionized by the Millipore Milli-Q system to the resistivity ≥ 18 MΩcm. Voltammetric measurements were carried out using the computer-controlled electrochemical system „PGSTAT 101“ (Eco-Chemie, Utrecht, The Netherlands), controlled by the electrochemical software “NOVA 1.5”. A three-electrode system (BioLogic, Claix, France) with glassy carbon electrode (GCE) of 3 mm in diameter as a working electrode, Ag/AgCl (3 mol/L NaCl) as a reference electrode and a platinum wire as a counter electrode were used. All potentials were expressed versus Ag/AgCl (3 mol/L NaCl) reference electrode. The supporting electrolyte (0.5 mol/L NaClO_4_), adjusted to the desired pH value, was placed in the electrochemical cell and the required aliquot of the standard analyte solution was added. The solution in the electrochemical cell was degassed with high purity nitrogen for 10 min before measurement, and the nitrogen blanket was maintained thereafter. Before each run, the glassy carbon working electrode was polished with diamond suspension in spray (grain size 6 µm) and rinsed with ethanol and deionized water. All experiments were performed at room temperature. The presented results are reported as the mean value of three independent measurements.

## 4. Conclusions

Novel purine, pyrrolo[2,3-*d*]pyrimidine, benzimidazole and indole derivatives with ferrocenylalkyl or ferrocenyl directly linked to *N*-1 of 1,2,3-triazole (**11a**−**11c**, **12a**−**12c**, **13a**−**13c**, **14a**−**14c**, **15a**−**15c**, **16a**, **23a**−**23c**, **24a**−**24c**, **25a**−**25c**) were prepared using Cu-promoted azide-alkyne cycloaddition (CuAAC) reaction of alkylated nitrogen heterocycles and azidoferrocene precursors. Purine and pyrrolo[2,3-*d*]pyrimidine with ferrocenyl attached to C-4 of 1,2,3-triazole (**34a**−**34c** and **35a**−**35c**) were obtained by CuAAC reaction of *N*-azidoethyl heterocycle derivatives and ethynyl-ferrocene. 6-Chloro-7-deazapurine **11a**−**11c** and 6-chloropurine **13a**, **15a** and **15b** ferrocenylalkyl derivatives exhibited pronounced cytostatic activities. Compounds **11c**, **13a** and **15b** found to be the most potent on colorectal adenocarcinoma (SW620) cells with IC_50_ of 9.07, 14.38 and 15.50 µM, respectively. However, preliminary assessment of permeability and metabolic stability in liver microsomes (mouse and human) showed that these compounds have a moderate to high permeability and are P-gp substrates. The 6-chloropurine moiety and *N*-7 regiosomerism are favored for metabolic stability although the overall instability was moderate to high, suggesting further optimization is required. Overall, *N*-9 and *N*-7 isomers of 6-chloropurine **13a** and **15a** containing ferrocenylmethylene unit are characterized with high solubility, good permeability and moderate stability in human liver microsomes. Although **15b** showed good physicochemical properties, this compound, together with **11a**−**11c**, is metabolically unstable. Voltammetric analysis showed that the oxidation of purine and pyrrolo[2,3-*d*]pyrimidine derivatives of ferrocene is influenced by the spacer between these moieties, showing that ferrocenyl group directly attached to *N*-1 of triazole in **11c** requires higher potential than **13a** and **15b** with ferrocene linked to triazole through an alkyl bridge, which may also contribute to their higher antioxidant properties.

Taken together, we may conclude that compound **13a** is highlighted for further structural optimization to obtain more effective and selective ferrocene-tagged purine and/or purine isostere derivatives against SW620 cell lines.

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
