# Peer review of "Purine and Purine Isostere Derivatives of Ferrocene: An Evaluation of ADME, Antitumor and Electrochemical Properties"

_molecules, 2020, doi:10.3390/molecules25071570_

Round 1

Reviewer 1 Report

The manuscript gives account on a complex synthetic and biological study on ferrocene conjugates with triazole linker. The research is well conducted and the results are presented adequatelly. Although the prepared compounds do not display outstanding cytotoxicity, the comprehensive physicochemical investigation including CV measurements provides valuable additional information about their drug-like properties. On the other hand, the Ru(II)-catalyzed synthesis and evaluation of the 1,5-disubstituted triazole-containing isomer of the most potent compound 11c are needed to improve the quality of this study establishing SAR and searching for more potent compounds. As for literature background, additional references should be cited in the Introduction reporting on the synthesis and anticancer effect of ferrocene conjugates with triazole linkers (e.g.: Bioorg & Med. Chem. Lett., 2016, 26, 946-949; Metallomics, 2017, 9, 1132-1141; J. Organomet. Chem. 2016, 813, 125-130).

Author Response

Reviewer #1

1. The manuscript gives account on a complex synthetic and biological study on ferrocene conjugates with triazole linker. The research is well conducted and the results are presented adequatelly. Although the prepared compounds do not display outstanding cytotoxicity, the comprehensive physicochemical investigation including CV measurements provides valuable additional information about their drug-like properties.

On the other hand, the Ru(II)-catalyzed synthesis and evaluation of the 1,5-disubstituted triazole-containing isomer of the most potent compound 11c are needed to improve the quality of this study establishing SAR and searching for more potent compounds.

The reviewer is right that, besides 1,4-disubstituted triazole-containing regioisomers, their 1,5-disubstituted isomers may have an influence on activity. The proposed extension of the presented study to additional synthesis and biological evaluation is time-consuming process and in this unprecedented situation due to current COVID-19 pandemic is hardly feasible, even though we are not convinced that 1,5-disubstituted triazole-containing isomer of the most potent compound 11c would ”improve the quality of this study establishing SAR”.

The aim of the presented study was to evaluate the series of 31 ferrocene-tagged purine and purine isosteres on antiproliferative activity with the aim to determine the impact of purine and its isosteres, and various 1,2,3-triazolylalkyl linker connected to ferrocene on their activity, ADME properties and electrochemical oxidation potential. However, the RuAAC synthesis and biological evaluation of a whole series of 1,5-disubstituted 1,2,3-triazolyl ferrocene−N-heterocycle hybrids are foreseen in the framework of research programme in order to obtain more comprehensive structure-activity relationship.

2. As for literature background, additional references should be cited in the Introduction reporting on the synthesis and anticancer effect of ferrocene conjugates with triazole linkers (e.g.: Bioorg & Med. Chem. Lett.,2016, 26, 946-949; Metallomics,2017, 9, 1132-1141; J. Organomet. Chem. 2016, 813, 125-130).

As suggested by reviewer, description of anticancer activity of ferrocene conjugates with triazole linkers along with indicated references were included in the Introduction (p. 2, lines 57-60) and References section (p. 23, lines 848-855).

Reviewer 2 Report

The contribution by Raic-Malic compiles a large number of valuable data, therefore the paper will be of interest for chemists and other scientists interested in bioinorganic and organometallic chemistry in the context of drug discovery.

The chemistry described is within the scope of known reactivities, therefore ma remarks mostly refer to formal and technical aspects:

  • The technical quality of all formula drawings should be checked. When I printed the manuscript larger pats were not readable; this is different when reading on a computer screen.
  • Scheme 2: Why are for X = N only 20-22 formed, and not the 1,3-double bond shifted isomers? The same applies to Scheme 3 (X = N=.
  • Line 115: Write chloro instead of chlorine.
  • Line 185: What are the products of metabolic degradation?
  • Line 258: What are the recommended drying agents? Reference?
  • No mass spectral data are given!
  • Line 506 and a number of other positions: Formula numbers should be printed in bold.
  • Line 807: Organomet. Chem.  (J. missing)

Author Response

Reviewer #2

1.The contribution by Raic-Malic compiles a large number of valuable data, therefore the paper will be of interest for chemists and other scientists interested in bioinorganic and organometallic chemistry in the context of drug discovery.

We appreciate the referee's comments on the value of our manuscript and are thankful for his/her encouraging words.

2. The chemistry described is within the scope of known reactivities, therefore my remarks mostly refer to formal and technical aspects:

The technical quality of all formula drawings should be checked. When I printed the manuscript larger pats were not readable; this is different when reading on a computer screen.

Formula drawings in Schemes 1-3 and Tables 1-3 have been rechecked and found to have good technical quality. There must be a discrepancy between the screen and print versions.

3. Scheme 2: Why are for X = N only20-22formed, and not the 1,3-double bond shifted isomers? The same applies to Scheme 3 (X = N).

Since 5-unsubstituted benzimidazole was used as a starting compound, the reaction of propargylation gave only 1-(propyn-1-yl) benzimidazole 20. In addition, indole and 5-iodoindole can give only N-1 propynyl indole 21 and 5-iodoindole 22 (Scheme 2). While N-propargylation of purines was carried out at elevated temperature (60 OC) for 24 h affording both N-9 (6-8) and N-7 (9, 10) regioisomers (Scheme 1), N-alkylation of 6-substituted purines was performed at room temperature for 24 h yielding only N-9 isomers of bromoethyl purines (31a-31c) (Scheme 3).

4. Line 115: Write chloro instead of chlorine.

'Chlorine' is replaced with 'chloro' (p. 4, line 118; p. 6, line 138).

5. Line 185: What are the products of metabolic degradation?

For evaluation of ADME properties, besides permeability and P-glycoprotein substrate assessment, metabolic stability parameters of selected compounds were assessed in human and mouse liver microsomes as described in 3.9. section. Products obtained by metabolic degradation were not characterized.

6. Line 258: What are the recommended drying agents? Reference?

Referring to the comment on drying agents, a sentence describing the preparation of anhydrous dimetylformamide (DMF) (p. 11, lines 262-265) and corresponding reference (p. 11, line 265; p. 25, lines 913, 914) were added. The methanol used for Cu(I)-catalysed azide-alkyne cycloadditions (CuAAC) was not dried. This is now corrected to be consistent throughout the manuscript (p. 2, line 75; p. 3, line 80; p. 4, line 105, p. 11, line 283; p. 18, line 578; p. 14, line 423).

7. No mass spectral data are given.

Instead of mass spectra, elemental analyses for novel compounds which is unequivocal support for the purity and homogeneity of compounds are provided.

8. Line 506 and a number of other positions: Formula numbers should be printed in bold.

All numbers designated to compounds are bolded now (Materials and Methods paragraph, lines 283, 423, 512, 513, 531, 532, 578).

9. Line 807:Organomet. Chem. (J. missing)

The accurate name of the journal is given (lines 816, 821). We apologize for this mistake.

Round 2

Reviewer 1 Report

The revised version of this manuscript can be accepted for publication in present form.